# Aβ1-42 and Tau as Potential Biomarkers for Diagnosis and Prognosis of Amyotrophic Lateral Sclerosis

**DOI:** 10.3390/ijms21082911

**Published:** 2020-04-21

**Authors:** Débora Lanznaster, Rudolf C. Hergesheimer, Salah Eddine Bakkouche, Stephane Beltran, Patrick Vourc’h, Christian R. Andres, Diane Dufour-Rainfray, Philippe Corcia, Hélène Blasco

**Affiliations:** 1UMR 1253, iBrain, University of Tours, Inserm, 37000 Tours, France; rchergesheimer@gmail.com (R.C.H.); patrick.vourch@univ-tours.fr (P.V.); Christian.andres@univ-tours.fr (C.R.A.); diane.dufour@univ-tours.fr (D.D.-R.); philippe.corcia@univ-tours.fr (P.C.); helene.blasco@univ-tours.fr (H.B.); 2Centre Constitutif SLA, CHRU Bretonneau, 37000 Tours, France; S.BAKKOUCHE@chu-tours.fr (S.E.B.); S.BELTRAN@chu-tours.fr (S.B.); 3CHU Tours, Service de Biochimie et Biologie Moléculaire, 37000 Tours, France; 4CHU Tours, Service de MNIV, 37000 Tours, France

**Keywords:** ALS, biomarker, Aβ1-42, total Tau, CSF

## Abstract

Amyotrophic lateral sclerosis (ALS) is the most common motor neuron disease, but its definitive diagnosis delays around 12 months. Although the research is highly active in the biomarker field, the absence of specific biomarkers for diagnosis contributes to this long delay. Another strategy of biomarker identification based on less specific but sensitive molecules may be of interest in clinical practice. For example, markers related to other neurodegenerative diseases such as Alzheimer’s disease (AD) could be fully explored. Here, we compared baseline levels of amyloidβ1-42 (Aβ1-42), total Tau, and phosphorylated-Tau (phospho-Tau) protein in the cerebrospinal fluid (CSF) of ALS patients to controls and correlated it with clinical parameters of ALS progression collected over 12 months. We observed increased levels of Aβ1-42 (controls: 992.9 ± 358.3 ng/L; ALS: 1277.0 ± 296.6 ng/L; *p* < 0.0001) and increased Aβ1-42/phospho-Tau ratio and Innotest Amyloid Tau Index (IATI) (both *p* < 0.0001). IATI and the phospho-Tau/total Tau ratio correlated positively with ALSFRS-R and weight at baseline. Multivariate analysis revealed that baseline ALSFRS-R was associated with Aβ1-42 and phospho-Tau/total Tau ratio (*p* = 0.0109 and *p* = 0.0013, respectively). Total Tau and phospho-Tau levels correlated negatively with ALSFRS-R variation at months 6 and 9, respectively (*p* = 0.02 and *p* = 0.04, respectively). Phospho-Tau/total Tau ratio correlated positively with ALSFRS-R variation at month 9 (*p* = 0.04). CSF levels of Aβ1-42 could be used as a complementary tool to ALS diagnosis, and total Tau and phospho-Tau levels may help establishing the prognosis of ALS. Further studies merit exploring the pathophysiological mechanisms associated with these markers. Despite their lack of specificity, phospho-Tau/total Tau and Aβ1-42 should be combined to other biological and clinical markers in order to improve ALS management.

## 1. Introduction

Amyotrophic lateral sclerosis (ALS) is a neurodegenerative disease characterized by the progressive loss of motor neurons and consequent progressive muscular paralysis. ALS patients usually die 3–5 years after the onset of symptoms by respiratory failure. Around 90% of ALS cases are sporadic, whereas the remaining cases are genetic forms [1]. ALS forms can be classified in bulbar and spinal, depending on the site of symptoms onset, with a higher severity in bulbar than in limb-onset patients. Currently, only two drugs are approved for the treatment of ALS (riluzole and edaravone), with moderate effects on survival or disease progression.

Clinically, ALS is a very diverse pathology, and definite diagnosis can be prolonged for up to 12 months, with this delay being in part due to the lack of diagnosis biomarkers. Biomarkers can also be applied to monitor disease progression or to follow treatment response. Although some biomarkers for ALS have been already proposed such as neurofilament and creatinine [2,3], none of them are validated in the clinical setting.

Concerning the diverse sensitive biomarkers admitted in some neurological diseases such as Alzheimer’s disease, the interest of their evaluation in ALS could be suggested. Despite their lack of specificity, they could represent an additional tool to help in ALS care. Recent studies have suggested a role of Tau protein as a biomarker for ALS diagnosis. Total Tau protein is a marker for neuronal lysis and is increased in the cerebrospinal fluid (CSF) of AD patients. Phosphorylated-Tau protein is associated with the pathophysiology of Alzheimer’s disease (AD), and is increased in AD as well [4,5,6,7]. In ALS, studies report opposite results [8,9,10,11,12], and some groups suggest that levels of CSF Tau at baseline could be correlated with disease progression [13,14,15]. Moreover, alterations in the amyloid processing protein (APP) and amyloid β1-42 peptide have been shown in the CSF and spinal cord of ALS patients [16,17,18]. Despite both the pathophysiological interest of AD markers in ALS and the reliability of their measurement, they have never been fully explored in a large cohort of ALS patients. In the present study, we investigated the diagnostic and prognostic role of amyloidβ1-42 (Aβ1-42), total Tau, and phosphorylated Tau proteins in the CSF of a large French cohort of ALS patients.

## 2. Results

### 2.1. Patients Characteristics

We analyzed data collected from 90 controls subjects and 123 ALS patients enrolled in the CHU of Tours, France (Table 1). Mean age was 67.45 years for controls and 66.06 years for ALS patients (*p* > 0.05). Males accounted for 63.4% of ALS patients vs. 56.7% in control subjects (*p* > 0.05). In total, 71% of ALS patients presented spinal onset. Mean disease duration from onset was 3.29 ± 2.03 years and from diagnosis was 2.04 ± 1.38 years. At baseline, mean score of ALS Functional Rating Scale-Revised (ALSFRS-R) was 39.77 ± 0.43, and we observed no difference between spinal onset (39.43 ± 0.54) and bulbar onset patients (40.93 ± 0.7). Baseline weight was 74.1+/− 1.4 kg and was not different according to the site of onset (spinal: 76.29 ± 1.67 kg; bulbar: 68.56 ± 2.30 kg).

### 2.2. Aβ1-42, Total Tau, and Phospho-Tau as Diagnosis Biomarkers in ALS

Baseline levels of β-amyloid (Aβ) peptide 1-42, as well as total and phosphorylated forms of Tau protein in the CSF from ALS patients and control subjects are presented in Table 2. We found increased levels of Aβ1-42 in the CSF of ALS patients compared to controls (ALS: 1277.07 ng/L, controls: 992.9 ng/L; *p* < 0.0001). Levels of total Tau did not differ between groups. Accordingly, these observations led to a different Innotest Amyloid Tau Index (IATI) index between controls and ALS patients (controls: 1.69 ± 0.62, ALS: 2.32 ± 0.66; *p* < 0.0001), and an increased ratio Aβ1-42/phosphorylated-Tau (phospho-Tau) (controls: 24.3 ± 11.6; ALS: 31.2 ± 9.8; *p* < 0.0001).

### 2.3. Correlation Between Biomarkers and Clinical Parameters at Baseline

ALSFRS-R score was correlated with disease duration from onset of symptoms (*p* = 0.043) and with age (*p* = 0.0013), but not with basal weight (*p* = 0.44). When analyzing the relationship between biomarkers and clinical parameters at baseline, we found many significant correlations (Table 3). More importantly, IATI and the phospho-Tau/total Tau ratio correlated with all parameters analyzed (Table 3, Figure 1). No differences were found when comparing biomarker levels between spinal-onset or bulbar-onset patients (data not shown). Multivariate analysis showed that baseline ALSFRS-R was linked with Aβ1-42 and phospho-Tau/total Tau ratio (*p* = 0.0109 and *p* = 0.0013, respectively).

### 2.4. Biomarkers for ALS Prognosis

Data were available from a smaller cohort of ALS patients including baseline data and data at months 6 (18 patients), 9, and 12 (17 patients). Total Tau levels correlated negatively with ALSFRS-R variation over 6 months (*p* = 0.0255), and phospho-Tau also correlated negatively with ALSFRS-R variation over 9 months (*p* = 0.0429). Phospho-Tau/total Tau ratio correlated positively with ALSFRS-R variation (*p* = 0.0419) over 9 months (Figure 2).

We also observed a trend for significance when analyzing the correlations of phospho-Tau and phospho-Tau/total Tau ratio with ALSFRS-R variation over 9 months (*p* = 0.06 and *p* = 0.07, respectively) (Table 4).

Aβ1-42 levels presented a trend of correlation with weight variation over 12 months (Spearman *r* = −0.46 [−0.77; 0.04], *p* = 0.065), but no other biomarker or ratios analyzed presented a significant correlation with weight variation at 6, 9, and 12 months (data not shown).

Fifty one patients (42%) from our cohort were dead at database lock. Mean survival of our cohort was 27.8 ± 32.3 (SD) months. None of the biological markers studied in this study were associated with survival (Cox proportional hazards model; Appendix A). In our cohort, ALSFRS-R at baseline also did not correlate with survival (HR: −0.008, 95% CI: −0.05; 0.04; *p* = 0.72).

## 3. Discussion

In this study, we showed that Aβ1-42 was increased in a large cohort of ALS patients in relation to control subjects, followed by an increase in the ratio Aβ1-42/P-tau and IATI index. We also showed that the phospho-Tau/total Tau ratio and IATI measured at diagnosis were positively correlated with ALSFRS-R and weight at baseline. Further, the phospho-Tau/total Tau ratio can be correlated with ALSFRS-R variation at different time points in the disease progression—the higher the ratio, the higher the variation in the ALSFRS-R score. To the best of our knowledge, this is the first time that this panel of biomarkers commonly used in the clinic to evaluate AD cases has been explored in a large cohort of ALS patients. The larger application of these biomarkers in the clinic and their high potential to discriminate among different neurodegenerative diseases [20,21,22], combined with more sensitive techniques, should improve the early diagnosis of ALS patients. Furthermore, investigation of the levels of such proteins in the CSF of ALS patients could also improve our understanding of the pathophysiology of this disease and shed a light in other pathological mechanisms that are thus far poorly investigated, as discussed below. They can also improve the discrimination between similar pathologies within a spectrum, as the ALS-frontotemporal lobar degeneration with tau or TDP-43 protein pathology [20].

### 3.1. Aβ1-42 Levels and Aβ1-42/Phospho-Tau Ratio as Diagnosis Biomarkers

Analysis of CSF in this ALS cohort revealed an increase in the levels of Aβ1-42 in comparison to control subjects. As a consequence, the IATI index (used to discriminate AD patients) was also higher in ALS patients than controls, as was the Aβ1-42/phospho-tau ratio. Previous studies reported contradictory results—although one study showed decreased levels of Aβ1-42 in the CSF of ALS patients [23], another found no difference between patients and controls [17]. The two main differences between these studies and ours is the cohort size (11, 12, and 123 ALS patients, respectively) and the disease duration—although patients in the study from Sjogren and co-authors [23] had a disease duration of 13.2 ± 4.5 years, our cohort presented a mean of 3.3 ± 2.0 years (Steinacker and co-authors [17] did not report the mean disease duration for ALS patients included in their study). Interestingly, a recent study also reported increased levels of Aβ1-42 in the CSF of ALS patients compared to controls subjects in a Chinese cohort [22]. Although not specific for ALS, some studies support a role for Aβ1-42 and amyloid precursor protein (APP) in the pathogenesis of neurodegenerative diseases than other AD, such as ALS [17,24]. Further, Aβ1-42 overexpression in rats induced accumulation of TDP-43, the hallmark of ALS in the cytoplasm of motor neurons [25], supporting the role of the amyloidogenic pathway in ALS pathogenesis. To increase the diagnostic specificity, these biomarkers could be combined with other imaging and clinical tests, thus improving diagnosis at early stages of the disease. In our cohort, we did not find any differences in the phosphorylated-Tau levels between control and ALS patients, as also reported before by Wilke et al. [12]. However, we know that these biomarkers lack specificity. Even so, we suggest that these biomarkers could be used as complementary tools for ALS diagnosis and prognosis. We also emphasize the need of a multi-centric study systematically investigating the levels and putative roles of such biomarkers in ALS patients as a way to reliably establish biomarkers to improve ALS clinical management.

### 3.2. Aβ1-42, Tau and Phospho-Tau Levels Were Related to Clinical Parameters at Baseline

Baseline correlations of biomarkers with disease parameters showed interesting results. At the time of diagnosis, Aβ1-42 levels were positively associated with the ALSFRS-R score (meaning that higher Aβ1-42 levels were associated with higher ASLFRS-R score), whereas total Tau and phospho-Tau were negatively correlated with basal weight (meaning that higher levels of Tau forms were associated with lower weight). Interestingly, the phospho-Tau/total Tau ratio was positively correlated with ALSFRS-R score and weight at baseline, as was the case for the IATI.

### 3.3. Mild Prognosis Role of Phospho-Tau and Total Tau in ALS

Regarding the levels of biomarkers and clinical parameters of disease progression, we found a negative correlation of phospho-Tau and total Tau with ALSFRS-R variation at months 6 and 9, respectively. We also found a positive correlation of phospho-Tau/total Tau ratio with ALSFRS-R variation at month 9. Except for a trend observed in the correlation of Aβ1-42 and weight variation at month 12, we did not find other correlations between the evaluated biomarkers and weight variation or survival. Conversely, another study showed that elevated levels of Aβ1-42 were associated with shorter survival [26]. A previous study found that total Tau was associated with shorter survival and higher progression rate, but not with ASLFRS-R [15]. Our study also reinforces previous findings showing that levels of total Tau, phospho-Tau, or the phospho-Tau/total Tau ratio were not correlated with survival [27].

As a retrospective study, one of the limitations of our findings is that we did not have follow-up data for all patients included in the baseline data analysis. Variation of ALSFRS-R and weight at months 6, 9, and 12 were systematically available only from around 15% of patients included in the baseline analysis (whereas survival data was available from all patients included in the study). A prospective multi-center study could confirm the prognostic role of AD biomarkers for ALS.

## 4. Materials and Methods 

### 4.1. Subjects

Data from 90 control subjects (subjects received in the Neurology Center of the Hospital of Tours with no neurological disease confirmed at the time of sampling) and 123 ALS patients with definite or probable ALS based on the revised El Escorial criteria were retrospectively collected. All patients were evaluated at the French ALS Center of Tours between October 2011 and May 2019. A sample of CSF was obtained at the first visit to analyze glucose, albumin, lactate, and total proteins as well as Aβ1-42, total Tau, and phosphorylated Tau. From the biomarker measurements, Innotest Amyloid Tau Index [28] was calculated for each patient. The IATI is an index used to discriminate AD patients: IATI = (measured Aβ1–42)/(240 + 1.18x measured total-Tau) [29,30]. Data such as age, sex, duration of symptoms, and site of onset were obtained at the first visit, as well as ALSFRS-R and weight. The age at onset was evaluated as the time at which motor weakness was first noted by the patient. Patients were monitored for weight and ALSFRS-R at 6, 9, and 12 months after disease onset. Survival was measured from time of diagnosis until death or database lock (December 2019).

### 4.2. Measurements

CSF was obtained by lumbar puncture following standard procedures, collected in a 15 mL polypropylene tube. Samples were centrifuged at 4000 × *g* for 10 min (4 °C) and immediately frozen at −80 °C until use. All CSF samples were analyzed for total Tau, phosphorylated Tau (phosphor-Tau), and amyloid-β42 (Aβ1-4242) by ELISA (INNOTEST, Fujirebio, Belgium), following the manufacturer’s instructions.

### 4.3. Statistical Analysis

In a first step, we evaluated the role of biological parameters as diagnosis biomarkers. Accordingly, we compared biological parameters between ALS patients and controls using a univariate analysis. All significant parameters that were independent from each other were included in a multivariate analysis. In a second step, we evaluated the relation between biological biomarkers and clinical parameters at baseline and all other time points. Then, we evaluated the prognosis role of the biological biomarkers. This analysis was based on the correlation between these biomarkers and the evolution of ALSFRS-R and weight at 6, 9, and 12 months (ALSFRS-R score at month of evaluation minus ALSFRS-R at baseline), and on survival analysis. Statistical analysis was performed using JMP 14.0VR (SAS Institute, Cary, North Carolina, USA). Univariate analysis was based on Mann–Whitney test or Student’s test for quantitative variables, and multivariate analysis was based on ANOVA test. Correlations were evaluated by Spearman’s test. Survival analysis was based on the Cox model. The level of significance was *p* < 0.05. Bonferroni adjustment was used for multiple tests. 

## 5. Conclusions

Here, we present important findings for the application of CSF biomarkers commonly used to diagnose Alzheimer’s disease towards an application for diagnosis and prognosis of ALS cases. We found in a big cohort of ALS patients that Aβ1-42 could be used to improve diagnosis, and our results suggest the putative role of phospho-Tau and total Tau protein in the CSF as markers for disease progression. We suggest that combination of these biomarkers with more specific, ALS-linked biomarkers such as TDP-43 in the CSF (as also suggested by Bourbouli, et al. [31]) will tremendously improve the clinical management of ALS patients.

## Figures and Tables

**Figure 1 ijms-21-02911-f001:**
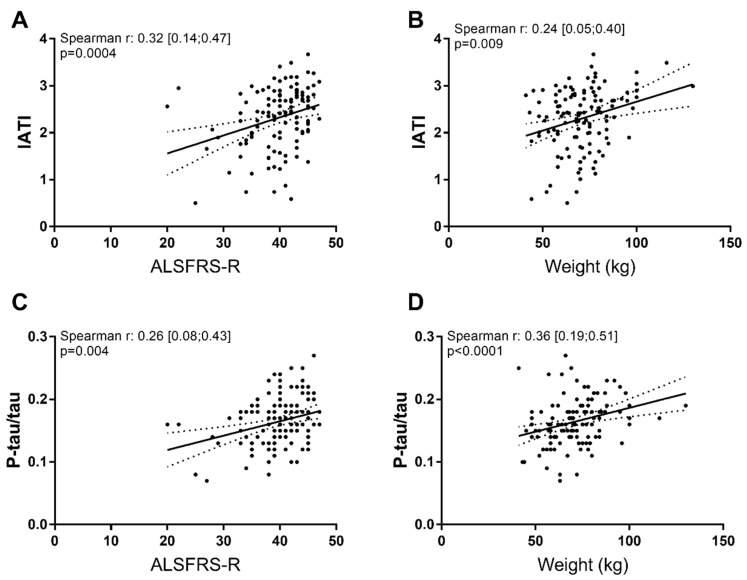
Significant correlations between biomarkers and clinical parameters at baseline. Innotest Amyloid Tau Index (IATI) (**A**,**B**) and ratio phosphorylated-Tau/total Tau (P-tau/tau) (**C**,**D**) correlated with ALSFRS-R and weight at baseline.

**Figure 2 ijms-21-02911-f002:**
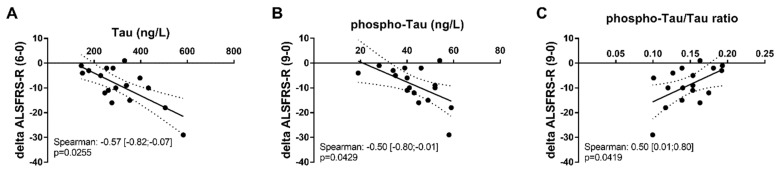
Correlation of ALSFRS-R variation with CSF biomarkers. (**A**) Correlation of total Tau with ALSFRS-R variation at month 6, (**B**) correlation of phospho-Tau with ALSFRS-R at month 9, and (**C**) correlation of phosphor-Tau/total Tau ratio with ALSFRS-R at month 9. Delta ALSFRS-R was calculated by subtracting the ALSFRS-R score at months 6 and 9 from ALSFRS-R at baseline.

**Table 1 ijms-21-02911-t001:** Characteristics of amyotrophic lateral sclerosis (ALS) patients and controls at the time of cerebrospinal fluid (CSF) collection.

	ALS	Control	*p*-Value
Number	123	90	
Gender (male)	63.41%	56.7%	0.33
Age (years)	66.06	67.45	0.35
Site of onset [19]	71%		
Disease duration (from onset; years)	3.29 ± 2.03		
Weigth at baseline (kg)	74.1 ± 1.4		
ALSFRS-R	39.77 ± 0.43		

Data is presented as mean ± standard deviation (SD).

**Table 2 ijms-21-02911-t002:** Biomarker levels in the CSF from controls and ALS patients.

Parameters	Controls	ALS	*p*
Aβ1-42 (ng/L)	992.9 ± 358.3	12,77.0 ± 296.6	**<0.0001**
Total Tau (ng/L)	485.3 ± 519.5	291.6 ± 140.6	0.18
Phospho-Tau (ng/L)	49.8 ± 27.9	44.5 ± 15.5	0.78
IATI	1.69 ± 0.62	2.32 ± 0.66	**<0.0001**
Ratio Aβ1-42/phospho-Tau	24.3 ± 11.6	31.2 ± 9.8	**<0.0001**
Ratio phospho-Tau/total Tau	0.15 ± 0.06	0.16 ± 0.38	0.48

Mean ± SD.

**Table 3 ijms-21-02911-t003:** Correlation between biomarkers and ALS clinical parameters at baseline.

Biomarker/Ratios	Basal ALSFRS-R	Basal Weight
Aβ1-42	**0.30 [0.12; 0.46]** ***p* = 0.001**	−0.47 [−0.23; 0.14] *p* = 0.61
Phospho-tau	0.05 [−0.14; 0.23] *p* = 0.59	**−0.21 [−0.38; −0.03]** ***p* = 0.02**
Total tau	−0.15 [−0.33; 0.03] *p* = 0.09	**−0.34 [−0.49; −0.17]** ***p* = 0.0001**
IATI	**0.32 [0.14; 0.47]** ***p* = 0.0004**	**0.24 [0.05; 0.40]** ***p* = 0.009**
Ratio Aβ1-42/P-tau	0.09 [−0.09; 0.28] *p* = 0.29	0.14 [−0.04; 0.32] *p* = 0.12
Ratio p-Tau/total Tau	**0.26 [0.08; 0.43]** ***p* = 0.004**	**0.36 [0.19; 0.51]** ***p* < 0.0001**

Values are given as Spearman correlation (*r*) with respective 95% confidence interval and *p*-value.

**Table 4 ijms-21-02911-t004:** Evaluation of biomarkers as prognostic for ALSFRS-R variation over 6, 9, and 12 months in relation to the baseline value of ALSFRS-R.

Parameter	Month 6 (*n* = 18)	Month 9 (*n* = 17)	Month 12 (*n* = 17)
Aβ1-42	−0.19 [−0.61; 0.32] *p* = 0.44	0.30 [−0.22; 0.69] *p* = 0.23	−0.36 [−0.72; 0.16] *p* = 0.16
Phospho-Tau	−0.46 [−0.77; 0.03] *p* = 0.06	**−0.50 [−0.80; 0.01]** ***p* = 0.04**	−0.17 [−0.61; 0.35] *p* = 0.50
Total Tau	**−0.54 [−0.82; −0.07]** ***p* = 0.02**	−0.24 [−0.65; 0.29] *p* = 0.36	−0.23 [−0.61; 0.23] *p* = 0.31
IATI	0.20 [−0.31; 0.62] *p* = 0.43	0.37 [−0.15; 0.73] *p* = 0.14	0.01 [−0.48; 0.50] *p* = 0.96
Ratio Aβ1-42/p-Tau	0.28 [−0.23; 0.67] *p* = 0.27	0.43 [−0.07; 0.76] *p* = 0.08	−0.05 [−0.45; 0.53] *p* = 0.85
p-Tau/total Tau	0.44 [−0.05; 0.76] *p* = 0.07	**0.50 [0.01; 0.80]** ***p* = 0.04**	0.22 [−0.31; 0.64] *p* = 0.40

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
