# Peer review of "Aβ1-42 and Tau as Potential Biomarkers for Diagnosis and Prognosis of Amyotrophic Lateral Sclerosis"

_ijms, 2020, doi:10.3390/ijms21082911_

Round 1
Reviewer 1 Report
The paper presented here claims about the role of CSF biomarkers for ALS diagnosis. Results like that should be carefully handled.
First, CSF biomarkers have been developed for AD, taking in account a pthophysiological mechanism involved in AD degeneration. I understand that a lot of research focused on these biomarkers for most of the neurodegenerative diseases with equivocal results however considering the last biomarker based classification such research should be differently oriented.
Second, reduction in Abeta or better AP mismetabolism would indicate a synaptic dysregulation, an increase is different and studies on pathological increases of Abeta are lacking.
Third, Total tau in controls appear moderately high.
Ethical issue: how authors obtained CSF from normal controls?
I think that despite the value of the manuscript, ALS certainly needs bimarkers for early diagnosis and likely for future pharmacological strategies, however I encourage authors to justify extensively the use of AD biomarkers in these cases.
Author Response
We are thankful for the comments made by the reviewer and we understand the raised concerns.
Individuals with neurological symptoms were received in the CHU of Tours and submitted to a battery of exams – including the obtention of CSF samples – to establish a diagnosis. As a part of this battery, the concentrations of tau, phospho-tau and beta amyloid peptides are evaluated to discard or confirm Alzheimer’s disease, including patients with probable ALS. Individuals without confirmed diagnosis for ALS, AD or other neurological diseases were grouped in the control group (as described in the Material and Methods, subsection 4.1, page 7: “Data from 90 control subjects (subjects received in the Neurology center of the Hospital of Tours with no neurological disease confirmed at the time of sampling)…”).
Common mechanisms of neurodegeneration were found for most of the neurodegenerative diseases (as axonal degeneration reflected in altered concentrations of tau and beta-amyloid). These common mechanisms, combined with the availability of these tests in the routine clinic prompted us (and other researchers) to investigate if such markers could be used at least to help in the early diagnosis of ALS – always combined with other biomarkers or clinical findings, as we suggest in our study.
Regarding Aβ1-42 increase, the decrease of Aβ1-42 concentrations in AD patients is related usually to the sequestration of beta amyloid peptides in the amyloid plaques, leading to the reduction of free Aβ peptides in the CSF (as reported by Grimmer et al. 2009. Beta Amyloid in Alzheimer’s Disease: Increased Deposition in Brain Is Reflected in Reduced Concentration in Cerebrospinal Fluid.). As in ALS cases there is no reporting of amyloid plaques containing Abeta peptides, it’s not impossible to find such free peptides in the CSF. In fact, another study was recently published reporting the same significant discriminating increase in Aβ1-42 concentrations in the CSF of ALS patients compared to controls and other neurodegenerative diseases (Ye et al., 2020. The Discriminative Capacity of CSF β-Amyloid 42 and Tau in Neurodegenerative Diseases in the Chinese Population. J Neurol Sci). We included this new reference in our manuscript to strengthen our findings.
Regarding tau concentrations, despite the moderate increase in CSF tau levels in our control subjects, it still falls into the normal range (100-500 ng/L), as described by Sjögren et al, 2001 (Tau and Aβ42 in Cerebrospinal Fluid from Healthy Adults 21–93 Years of Age: Establishment of Reference Values. Clinical Chemistry).
Finally, we want to emphasize that our study brings new data about the use of biomarkers commonly used in AD that can be used also for ALS, but always in combination with other more specific biomarkers and clinical investigation. Finally, other studies with larger cohorts should be conducted to confirm our findings, as emphasized in our manuscript (Discussion section, page 6).
Reviewer 2 Report
The goal of the paper by Lanznaster et al. is widely appreciable given the current absence of specific biomarkers in the diagnosis of amyotrophic sclerosis.
One of the main needs to which research is attempting to identify and slow down the course of diseases, such as amiotrophic lateral sclerosis, is the study of molecules that predict the pathological form, hence the need to find specific biomarkers capable of identifying subjects potentially at risk of developing this pathology.
I pay attention to a particular marker studied here. Tau represents a fundamental molecule in this field of study. In humans, pathological hyperphosphorylation of tau leads to microtubule destabilization and formation of neurofibrillary tangles, one of the hallmarks of Alzheimer’s disease and other Tauopathies (see Bakota and Brandt 2016, Tau biology and tau-directed therapies for Alzheimer’s disease. Drugs 76:301–313; Arendt, T., Stieler, J., Holzer, M. Brain hypometabolism triggers PHF-like phosphorylation of tau, a major hallmark of Alzheimer’s disease pathology. J. Neural Transm. 2015, 122, 531–539.).
The paper deserves to be published, I suggest mentioning more recent publications such as those above cited.
Author Response
We appreciate all the positive comments made by the reviewer. We included the more recent references suggested by the reviewer in this new version of our manuscript (Introduction, page 2, lines 57-58).